# Primary decidual zone formation requires Scribble for pregnancy success in mice

Jia Yuan [1,2,6], Shizu Aikawa [1,2,6], Wenbo Deng[1,2,5,6], Amanda Bartos[1,2], Gerd Walz [3], Florian Grahammer[4], Tobias B. Huber[4], Xiaofei Sun [1,2] & Sudhansu K. Dey [1,2]*

Scribble (Scrib) is a scaffold protein with multifunctional roles in PCP, tight junction and Hippo signaling. This study shows that Scrib is expressed in stromal cells around the implantation chamber following implantation. Stromal cells transform into epithelial-like cells to form the avascular primary decidual zone (PDZ) around the implantation chamber (crypt). The PDZ creates a permeability barrier around the crypt restricting immune cells and harmful agents from maternal circulation to protect embryonic health. The mechanism underlying PDZ formation is not yet known. We found that uterine deletion of *Scrib* by a *Pgr-Cre* driver leads to defective PDZ formation and implantation chamber (crypt) formation, compromising pregnancy success. Interestingly, epithelial-specific *Scrib* deletion by a *lactoferrin-Cre* (*Ltf-Cre*) driver does not adversely affect PDZ formation and pregnancy success. These findings provide evidence for a previously unknown function of stromal Scrib in PDZ formation, potentially involving ZO-1 and Hippo signaling.

---

[1] Division of Reproductive Sciences, Cincinnati Children's Hospital Medical Center, Cincinnati, OH, USA. [2] College of Medicine, University of Cincinnati, Cincinnati, OH, USA. [3] Department of Medicine, Renal Division, Medical Center-University of Freiburg, Faculty of Medicine, University of Freiburg, Freiburg, Germany. [4] Department of Medicine III, University Medical Center Hamburg-Eppendorf, Hamburg, Germany. [5] Present address: Fujian Provincial Key Laboratory of Reproductive Health Research, School of Medicine, Xiamen University, Xiamen, Fujian, China. [6] These authors contributed equally: Jia Yuan, Shizu Aikawa, Wenbo Deng   *email: sk.dey@cchmc.org

The uterus shows dramatic morphological and molecular changes involving an interplay of ovarian hormones progesterone ($P_4$) and estrogen ($E_2$) during blastocyst implantation, a gateway to pregnancy success. In mice, embryos from the oviduct begin to enter the uterus late on day 3 and complete their journey on the morning of day 4 (day 1 = vaginal plug). In the evening of day 4, the blastocyst initiates regularly spaced attachment with the luminal epithelium (attachment reaction), followed by homing into a spear-shaped crypt that emerges with glands on day 5[1,2].

The reciprocal communication between an implantation-competent blastocyst and a receptive uterus is central to the implantation process and pregnancy success. A defect in any aspect of this process results in implantation failure or propagates adverse ripple effects for the remainder of gestation. We have recently shown a direct communication between the glands and the implanting embryo within the crypt (implantation chamber). The Wnt5a-ROR-PCP (planar cell polarity) signaling pathway is a major contributor to the architecture of this gland-crypt landscape[3,4], in which aberrant signaling in the uterus causes severely compromised pregnancy outcomes due to faulty implantation[2].

One of the early events following implantation is stromal cell differentiation into a specialized type of cells termed decidual cells (decidualization) that support embryo development. In mice and rats, this process is initiated at the antimesometrial (AM) pole with the transformation of stromal cells into epithelial-like cells (epithelioid cells) surrounding the epithelium of the crypt (implantation chamber) that houses the blastocyst. This zone is called the primary decidual zone (PDZ), first identified by Krehbiel[5] as an avascular zone and thought to function as a transient, size-dependent permeable barrier to protect the embryo from harmful agents, such as immunoglobulins, immune cells, microorganisms, and other noxious agents[6]. This zone begins to form from day 5 afternoon and becomes fully established on day 6 with the loss of the crypt epithelium[7]. The PDZ gradually undergoes demise with the emergence of the secondary decidual zone (SDZ), a proliferating and differentiating stromal cell layer around the degenerating PDZ that peaks on day 8 in mice. The PDZ is considered the first line of defence to safeguard the implanting embryo as it transitions from its epithelial to stromal residence.

Scrib is a multifunctional scaffold protein involved in cell polarity, cell adhesion, and cell proliferation[8]. As an important component of the PCP complex, Scrib interacts with Vangl2 and contributes to embryo development, organogenesis, and tumorigenesis[9,10]. Our previous study revealed that Scrib displays spatial and temporal expression patterns in the epithelium and stroma during early stages of pregnancy: it shows distinct expression in the apical surface of the epithelium and low expression in the stromal cells prior to embryo implantation[4]. Following blastocyst attachment, Scrib ceases to express in the luminal epithelium with distinct expression in the stroma surrounding the implantation chamber[4].

Here we show that Scrib is critical for initiating PDZ formation. Scrib is perhaps not involved in epithelial cell polarization in early stages of pregnancy, since conditional deletion of Scrib in the uterine epithelium by a Ltf-Cre driver (Scrib^{f/f}Ltf^{cre/+}) shows normal pregnancy outcomes. Furthermore, tri-dimensional (3D) visualization of implantation sites after tissue clearing displays comparable gland-implantation chamber assembly in Scrib^{f/f} Ltf^{cre/+} mice. In contrast, Scrib deletion in the uterine stromal cells by a Pgr-Cre driver (Scrib^{f/f}Pgr^{cre/+}) manifests aberrant crypt-glands association with implantation defects, resulting in adverse pregnancy outcomes. More importantly, we found that Scrib activates the Hippo signaling pathway, inhibiting

proliferation of stromal cells underlying the implantation chamber to initiate the PDZ formation. This is a previously unknown function of Scrib in stromal cell remodeling to form the avascular PDZ, a critical step in protecting the embryo from harmful infiltrators.

## Results

**Stromal deletion of Scrib impedes pregnancy outcomes.** Scrib can interact with Vangl2, Dlg, and Lgl to execute PCP signaling[8]. Our previous study shows that Scrib has a distinct expression pattern at the apical surface of uterine epithelial cells in early stages of pregnancy followed by increasing expression in stromal cells around the blastocyst at the implantation site[4]. Uterine ablation of Vangl2 causes aberrant expression of Scrib in both the epithelium and stroma[2,4]. To assess the function of Scrib during early pregnancy, we generated two mouse lines: mice with uterine deletion of Scrib using a Pgr-Cre/+ driver, which can delete genes of interest in the epithelium, stroma, and myometrium[11], and mice with deletion through a Ltf-Cre driver, which can delete genes primarily in the epithelium[12]. Indeed, immunofluorescence (IF) staining using an anti-Scrib antibody shows effective deletion of Scrib in the expected cell types in pregnant uteri on days 4, 5, and 6 by each of the Cre-drivers (Fig. 1a and Supplementary Fig. 1). In day 5 pregnant Scrib^{f/f}Ltf^{cre/+} mice, Scrib expression in stromal cells surrounding the blastocyst is clearly seen (Fig. 1a).

Upon examination of pregnancy outcomes in Scrib-deleted females, we found that litter sizes are comparable between the floxed and Scrib^{f/f}Ltf^{cre/+} mice (Fig. 1b), albeit with somewhat lower pregnancy success rate (Fig. 1b and Supplementary Table 1). In contrast, pregnancy outcomes in Scrib^{f/f}Pgr^{cre/+} females were significantly inferior: only 43% of plug-positive Scrib^{f/f}Pgr^{cre/+} dams gave birth to smaller litter sizes as compared with Scrib^{f/f} and Scrib^{f/f}Ltf^{cre/+} females. The remaining 57% failed to deliver any live pups (Fig. 1b and Supplementary Table 1). With these results in hand, we assessed the state of pregnancy on day 12 and found numerous resorption sites in Scrib^{f/f}Pgr^{cre/+} females (Fig. 1c). To trace back the initiation of adverse effects, we found the beginning of embryo demise within significantly smaller implantation sites on day 8 of pregnancy in Scrib^{f/f}Pgr^{cre/+} mice (Fig. 1d). Conversely, no significant adverse effects were noted in Scrib^{f/f}Ltf^{cre/+} mice (Fig. 1d). These results provide evidence that stromal Scrib is critical for pregnancy success, although the role of both stromal and epithelial Scrib cannot be completely ruled out in Scrib^{f/f}Pgr^{cre/+} females, since Pgr-Cre can delete Scrib in both cell types. Notably, distinct signals of Scrib are observed in uterine blood vessels in the stromal bed (Fig. 1a). Pgr-Cre is incapable of deleting genes of interest in endothelial cells, since progesterone receptor (Pgr) is not expressed in these cells[13]. Therefore, Scrib function in uterine endothelial cells remains unknown at this time.

To further explore the effects of uterine Scrib deletion that impair pregnancy outcomes in Scrib^{f/f}Pgr^{cre/+} females, we assessed the expression of Hbegf and Ptgs2 (encoding Cox2), two important markers of the attachment reaction during implantation[1]. We found that their expression patterns and timing are comparable in both Scrib^{f/f}Pgr^{cre/+} and Scrib^{f/f}Ltf^{cre/+} females compared with floxed females (Fig. 1e). In the same context, expression patterns of Pgr and estrogen receptor (Esr1), two primary mediators of $P_4$ and $E_2$ actions in the uterus, are comparable between floxed and Scrib-deleted uteri on day 4 morning (Supplementary Fig. 2). We also assessed stromal cell proliferation on day 4, which is critical for implantation and the ensuing decidualization[1,14]. Immunolocalization of Ki67 and pHH3 in day 4 uteri are comparable among mice of three genotypes, suggesting a normal uterine state of receptivity in Scrib^{f/f}, Scrib^{f/f}Ltf^{cre/+}, and Scrib^{f/f}Pgr^{cre/+} mice (Supplementary

Fig. 3). Taken together, these results show a previously unrecognized role of Scrib in sub-epithelial stromal cells cooperating with the epithelium to construct a unique gland-implantation chamber to facilitate embryo attachment.

**Stromal deletion of *Scrib* downregulates decidual marker gene expression and compromises implantation chamber formation**. The observations of abnormal pregnancy outcomes in *Scrib^f/fPgr^cre/+*, but apparently not in *Scrib^f/fLtf^cre/+*, led us to investigate the morphological landscapes of these *Scrib*-deleted uteri during embryo implantation. Histological analysis shows that the shape of the implantation chamber on day 5 morning is aberrant in *Scrib^f/fPgr^cre/+* mice (Fig. 2a). Our recent studies have shown that the formation of an appropriately shaped crypt chamber is key to implantation success[2,4]. Therefore, we examined the crypt-gland landscape after tissue clearing and captured 3D images of day 5 pregnancy in floxed, *Scrib^f/fPgr^cre/+* and *Scrib^f/fLtf^cre/+* mice. We found that *Scrib* deletion in the epithelium by *Ltf-Cre* driver does not significantly alter the crypt-gland assembly (Fig. 2b), albeit some glands were underdeveloped. In contrast, the typical landscape of crypt-gland architecture is absent or aberrant in *Scrib^f/fPgr^cre/+* mice (Fig. 2b). Furthermore, the crypt epithelium of

*Scrib^f/fPgr^cre/+* uteri fails to remodel as a spear-shaped structure seen in floxed and *Scrib^f/fLtf^cre/+* mice (Fig. 2).

We also examined the expression of a well-known decidual marker *Bmp2* on day 5[15,16]. Normally, *Bmp2* begins to express in the stromal cells surrounding the embryo at the implantation site after the attachment reaction and gradually increases in these cells during the progression of pregnancy[16]. We found *Bmp2* expression is substantially lower at the implantation site in *Scrib^f/fPgr^cre/+* mice on day 5, suggesting aberrant initiation of decidualization (Fig. 3a). These experiments were followed by localization of *Wnt4*, another marker of decidualization[17,18]. *Wnt4* is expressed at very low levels during the preimplantation period but begins to express in stromal cells surrounding the blastocyst after attaching to the uterus[17]. Our in situ hybridization results show very low levels of *Wnt4* expression in stromal cells surrounding the blastocyst in *Scrib^f/fPgr^cre/+* uteri as compared with floxed or *Scrib^f/fLtf^cre/+* mice (Fig. 3a). *Hoxa10* mutant uteri show abnormal *Wnt4* expression in the stroma, suggesting that *Wnt4* is a downstream target of *Hoxa10*[17]. *Hoxa10* is expressed in stromal cells during the receptive phase (day 4) with more intense expression with the initiation of decidualization[7,19]. We found depressed *Hoxa10* expression in stromal cells in day 5 implantation sites of *Scrib^f/fPgr^cre/+* mice

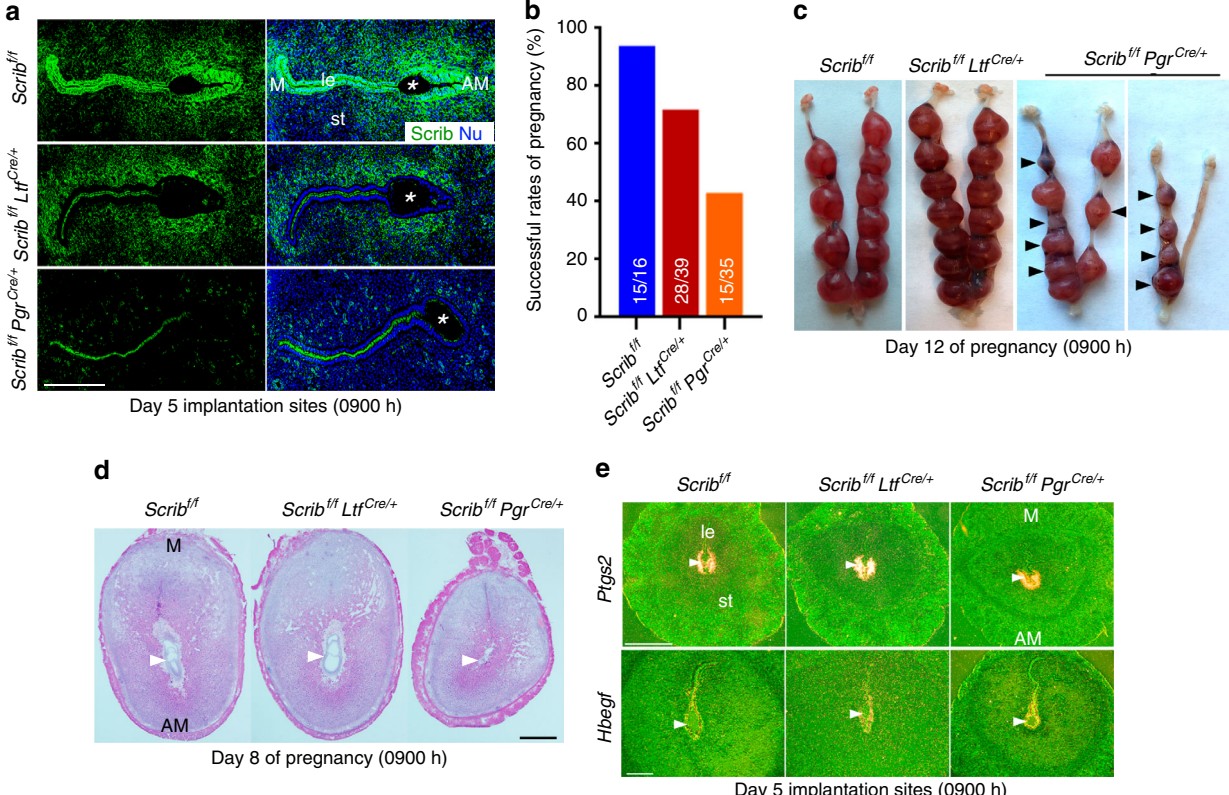

**Fig. 1** Disparate pregnancy outcomes in mice deleted of Scrib in the uterus by *Pgr-Cre* and *Ltf-Cre*. Uterine ablation of Scrib results in compromised pregnancy outcomes, but exclusively deleted Scrib in uterine epithelium displays normal fertility. **a** IF localization of Scrib in implantation site on day 5 of pregnancy in each genotype mouse. Asterisks indicate the location of the blastocysts. Scale bar, 200 μm. **b** Pregnancy success rate of *Scrib^f/f*, *Scrib^f/fLtf^cre*, and *Scrib^f/fPgr^cre/+* females. Numbers within the bars indicate the number of females that gave birth to live pups compared with the total number of plug-positive females in each genotype. A number of females were bred more than once contributing to the increased number of mice examined for pregnancy outcomes. **c** Representative images of day 12 implantation sites in *Scrib^f/f*, *Scrib^f/fLtf^cre/+*, and *Scrib^f/fPgr^cre/+* females. Arrowheads indicate embryo resorption sites. **d** Histology of day 8 implantation sites in each genotype, showing reduced size of the decidual area with degenerating embryos in *Scrib^f/fPgr^cre/+* females. Arrowheads indicate the location of embryos. Scale bar, 500 μm. **e** In situ hybridization of *Ptgs2* and *Hbegf* in *Scrib^f/f*, *Scrib^f/fLtf^cre/+*, and *Scrib^f/fPgr^cre/+* mice in implantation sites on day 5 of pregnancy. Arrowheads indicate the location of blastocysts. Scale bar, 500 μm (upper panels) and 200 μm (lower panels). Each image is a representative of at least three independent experiments. M mesometrial pole, AM antimesometrial pole, le luminal epithelium, st stroma.

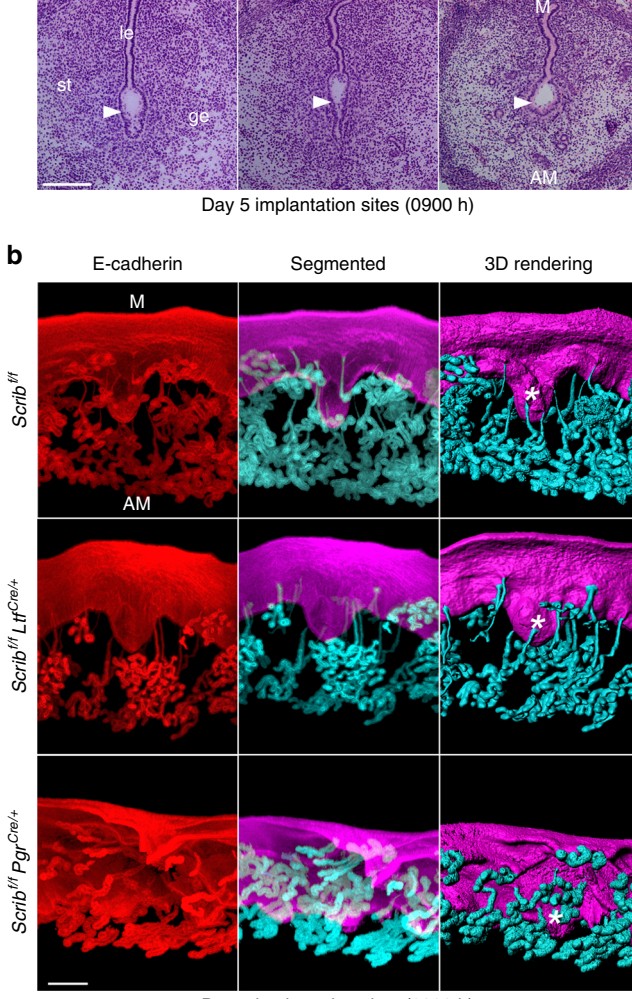

**Fig. 2** Implantation is aberrant in *Scrib^{f/f}Pgr^{cre/+}* mice. **a** Histology of day 5 of implantation sites in each genotype. Scale bar, 200 μm. Arrowheads indicate the location of embryos. **b** 3D imaging of day 5 implantation sites in *Scrib^{f/f}*, *Scrib^{f/f}Ltf^{cre/+}*, and *Scrib^{f/f}Pgr^{cre/+}* females. Images of E-cadherin immunostaining, segmented, and 3D rendered images of day 5 implantation sites in each genotype show the aberrant crypt-gland structure in *Scrib^{f/f}Pgr^{cre/+}* females. Images were generated by a Nikon A1R Multiphoton Microscope with LWD 16× objective with 3 μm Z-stack. Scale bar, 200 μm. Asterisks indicate the location of blastocysts. Each image is a representative of at least three independent experiments. M mesometrial pole, AM antimesometrial pole, le luminal epithelium, ge glandular epithelium, st stroma.

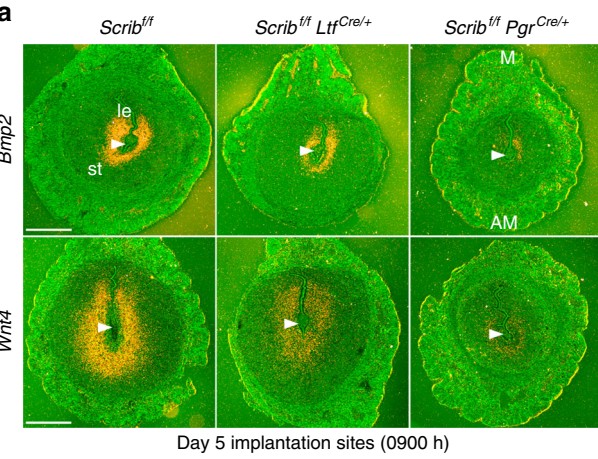

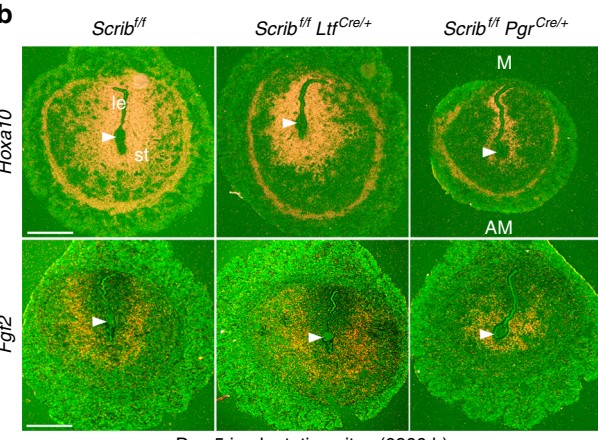

**Fig. 3** Stromal cell *Scrib* deletion compromises initiation of decidualization. **a**, **b** In situ hybridization of *Bmp2, Wnt4, Hoxa10,* and *Fgf2* in each genotype on day 5 implantation sites show robust disrupted pattern in *Scrib^{f/f}Pgr^{cre/+}* mice. Scale bar, 500 μm. Arrowheads indicate the location of blastocysts. Each image is a representative of at least three independent experiments. M mesometrial pole, AM antimesometrial pole, le luminal epithelium, st stroma.

(Fig. 3b). *Fgf2* is also expressed in stromal cells around the blastocyst in a similar fashion as *Bmp2* and *Wnt4* on day 5[16]. In contrast to the expression of *Bmp2 and Wnt4*, the expression of *Fgf2* is distinct in stromal cells surrounding the blastocyst in *Scrib^{f/f}Pgr^{cre/+}* mice at the area destined to become the PDZ in floxed uteri (Fig. 3b). Since *Fgf2* is known to regulate cell proliferation[20], we examined stromal cell proliferation on day 5 by Ki67 and pHH3 immunostaining. We found no alteration of epithelial cell proliferation, but stromal cells surrounding the implanting blastocyst show intense cell proliferation in *Scrib^{f/f}Pgr^{cre/+}* mice (Supplementary Fig. 4). By comparison, stromal cells surrounding the embryo in floxed and *Scrib^{f/f}Ltf^{cre/+}* mice show reduced cell proliferation, suggesting that initiation of differentiation of these cells promotes the formation of PDZ.

Stromal cell proliferation and differentiation during decidualization is governed by a complex interplay of transcriptional factors, cytokines, and cell differentiation genes[1]. In rodents, stromal cells surrounding the blastocyst cease proliferation and undergo differentiation to form the PDZ, which is comprised of epithelioid-like cells[7,21]. $P_4$-Pgr signaling is indispensable for decidualization[22]. We found Pgr expression in the stroma surrounding the implantation chamber in day 5 *Scrib^{f/f}* and *Scrib^{f/f}Ltf^{cre/+}* uteri (Supplementary Fig. 4c). In contrast, Pgr expression in these cells is substantially reduced in *Scrib^{f/f}Pgr^{cre/+}* mice (Supplementary Fig. 4c), although comparable Pgr expression is seen in uteri of each genotype on day 4 (Supplementary Fig. 2a). Collectively, our results provide evidence that *Scrib* deletion in stromal cells begins to impair stromal cell transformation to decidual cells starting from day 5 afternoon.

**Scrib deletion leads to aberrant Hippo signaling and dysregulates PDZ formation.** One important role of Scrib is to regulate cell proliferation involving the Hippo signaling pathway[8,23]. The Hippo signaling pathway is a key regulator of organ size and tumorigenesis and is primarily comprised of mammalian Mst1/2, Lats1/2, Sav1, and YAP[24,25]. The kinase activities executed by Mst1/2 and Lats1/2 are core components of the Hippo signaling

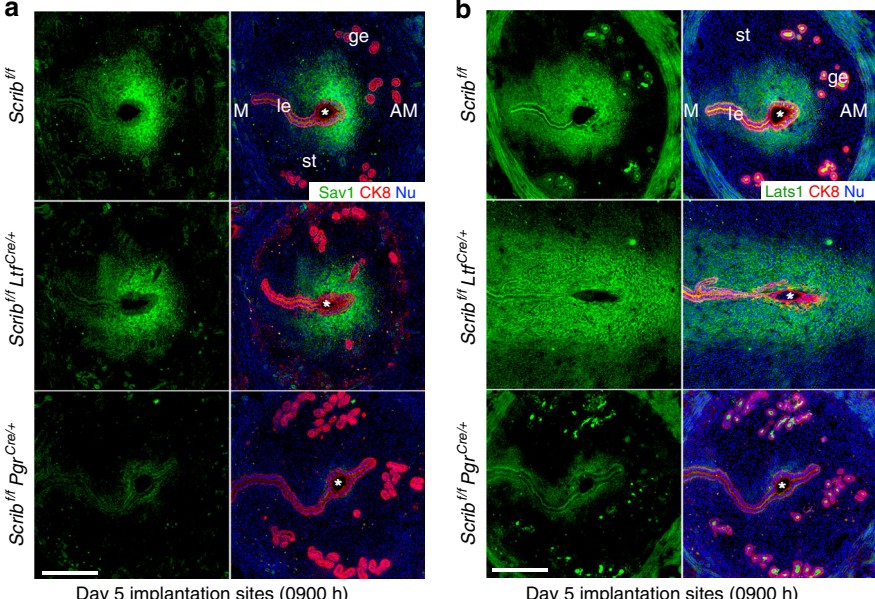

**Fig. 4** Hippo signaling is inactive on day 5 morning implantation sites of $Scrib^{f/f}Pgr^{cre/+}$ females. **a, b** IF of Sav1 and Lats1 on day 5 morning implantation sites of $Scrib^{f/f}$, $Scrib^{f/f}Ltf^{cre/+}$, and $Scrib^{f/f}Pgr^{cre/+}$ mice. Scale bar: 200 μm. Asterisks indicate the location of blastocysts. Each image is a representative of at least three independent experiments. M mesometrial pole, AM antimesometrial pole, le luminal epithelium, st stroma, ge glandular epithelium.

pathway. In response to intracellular and extracellular stimuli, Mst1/2 can phosphorylate Lats1/2 with Sav1 as an adaptor protein[26]. Phosphorylated Lats1/2 then executes kinase activities on YAP to suppress its nuclear translocation[27]. Since YAP works as a co-transcriptional factor of TAZ to facilitate cell proliferation, the activated Mst1/2-Sav1-Lats1/2 axis functions as an inhibitor for the transcriptional activity of YAP–TAZ[25]. There is evidence that Scrib attenuates cell proliferation via YAP–TAZ by interacting with Mst1/2 and Lats1/2 to activate their kinase activities[8,28]. This finding in conjunction with our observation of aberrant cell proliferation in $Scrib^{f/f}Pgr^{cre/+}$ uteri suggests that Hippo signaling is affected. In this respect, we found that Sav1 and Lats1 are expressed in the stroma surrounding the implantation site with similar expression patterns to Scrib in $Scrib^{f/f}$ and $Scrib^{f/f}Ltf^{cre/+}$ uteri on day 5 morning (Fig. 4). However, their expressions are much weaker in the stroma of $Scrib^{f/f}Pgr^{cre/+}$ uteri (Fig. 4). These results suggest that activation of the Hippo signaling pathway is compromised in the absence of stromal Scrib.

As previously stated, Scrib expression is suppressed in the crypt epithelium with simultaneous upregulation in underlying stroma cells after embryo attachment; the expression becomes stronger in stromal cells at the implantation sites in the afternoon of day 5 compared with day 5 morning (Fig. 5a). In mice, the PDZ begins to form on day 5 afternoon and becomes fully established on day 6 morning with termination of cell proliferation[1,7]. The observation of sustained cell proliferation (Supplementary Fig. 4a, b) and downregulation of expression of decidual markers on day 5 morning (Fig. 3a) in the stroma led us to ask if PDZ formation becomes dysregulated in $Scrib^{f/f}Pgr^{cre/+}$ mice. As stated above, ZO-1 is an epithelial and PDZ marker in WT mice[21]. We observed ZO-1 expression in the epithelium and blood vessels, but it is absent or very low in stromal cells at the implantation sites of each genotype on day 5 morning, since the PDZ is yet to be formed (Supplementary Fig. 4d). In contrast, ZO-1 is expressed in the underlying stromal cells at the implantation site in $Scrib^{f/f}$ and $Scrib^{f/f}Ltf^{cre/+}$ mice on day 5 afternoon but is absent in $Scrib^{f/f}Pgr^{cre/+}$ females, indicating that PDZ formation of $Scrib^{f/f}Pgr^{cre/+}$ females is interrupted (Fig. 5a). These results suggest that stromal Scrib is critical for PDZ formation. One of

the established roles of Scrib is maintenance of junctional networks. Importantly, Scrib has been shown to bind to ZO-1[29]. Deletion or downregulation of Scrib results in impaired cell-cell junctions[8]. These contexts also support our conjecture that Scrib stabilizes tight junctions by ZO-1 and contributes to the formation of PDZs.

We also examined the expression of Sav1 and Lats1 in the uterus on day 5 afternoon. Both display stronger expression in stromal cells underlying the embryo on day 5 afternoon compared with day 5 morning with partial overlap with ZO-1 expression in $Scrib^{f/f}$ and $Scrib^{f/f}Ltf^{cre/+}$ females (Fig. 5b, c). Their expression is notably lower in $Scrib^{f/f}Pgr^{cre/+}$ mice (Fig. 5b, c). It is also noteworthy that the expressions of Scrib, Sav1, and Lats1 are wider and appear earlier than ZO-1 in stromal cells surrounding the implantation chamber. The increased Hippo signaling restricts YAP–TAZ activity[8,30]. Our findings of distinct expression of YAP in the designated PDZ area in $Scrib^{f/f}Pgr^{cre/+}$ uteri suggest that Scrib initiates PDZ formation by turning on the Hippo signaling pathway (Fig. 5d).

**Stromal cell deletion of *Scrib* fails to establish the PDZ.** We examined uterine morphology on day 6 of pregnancy when PDZ formation is normally fully established[7]; this process is initiated on day 5 afternoon. We consistently found that $Scrib^{f/f}Pgr^{cre/+}$ mice show smaller implantation sites on day 6 due to the failure of PDZ formation (Fig. 6a). *Bmp2* is a marker gene for decidualization, expressed in stromal cells surrounding the implanting embryo with the initiation of decidualization followed by expanded expression on day 8[16]. Indeed, we observed robust expression of *Bmp2* in floxed and $Scrib^{f/f}Ltf^{cre/+}$ decidua on day 6 (Fig. 6b) compared with that seen on day 5 (Fig. 3a). In contrast, *Bmp2* expression is absent in stromal cells close to the implanting embryo in $Scrib^{f/f}Pgr^{cre/+}$ uteri on day 6 (Fig. 6b), providing evidence for compromised decidualization in these mice.

In mice, PDZ formation is associated with cessation of cell proliferation[7]. In $Scrib^{f/f}Pgr^{cre/+}$ mice, cell proliferation as marked by Ki67 immunostaining continues in the PDZ even on day 6 (Fig. 6c), suggesting aberrant PDZ formation in the absence of

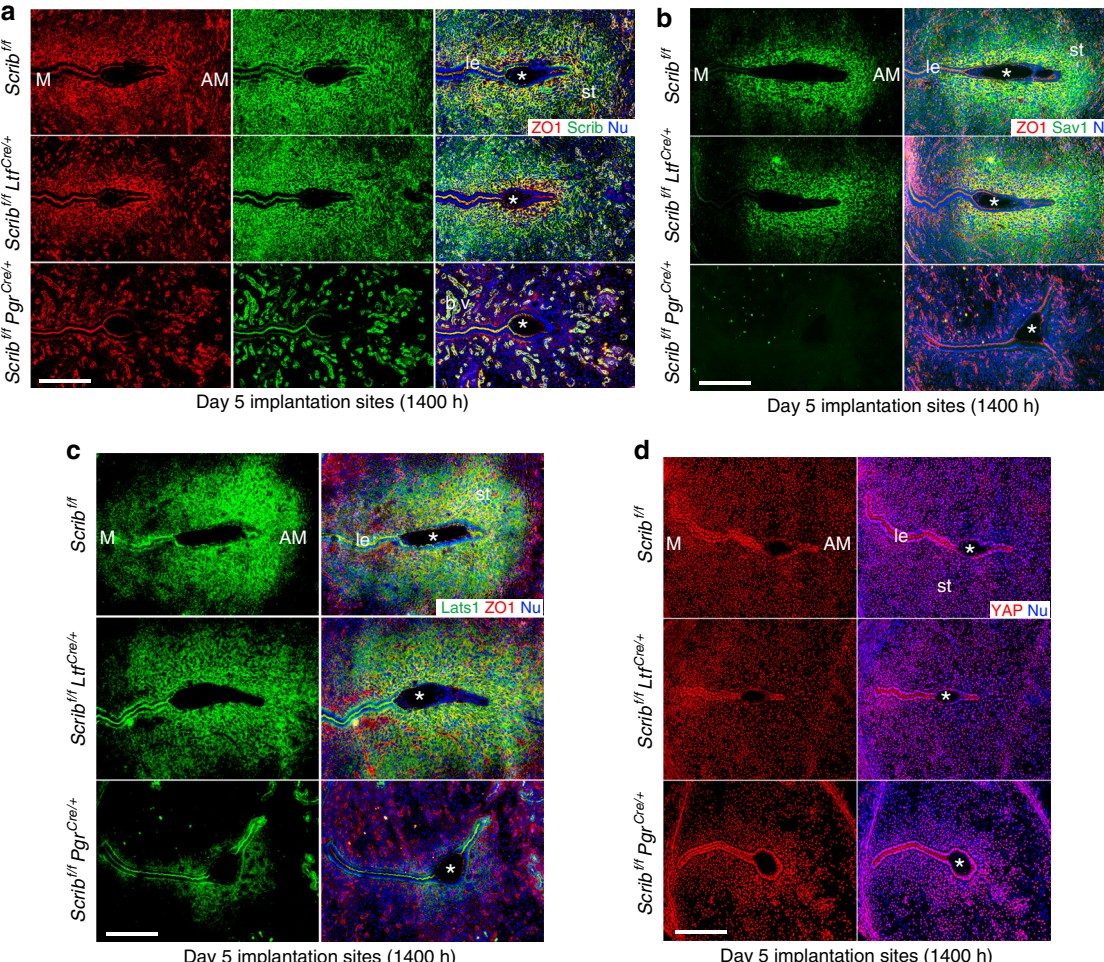

**Fig. 5** On-time initiation of PDZ is derailed in *Scrib^{f/f}Pgr^{cre/+}* uteri. **a** IF co-localization of ZO-1 and Scrib in *Scrib^{f/f}*, *Scrib^{f/f}Ltf^{cre/+}*, and *Scrib^{f/f}Pgr^{cre/+}* mice on day 5 afternoon (1400 h) implantation sites. PDZ represented by ZO-1 is absent in *Scrib^{f/f}Pgr^{cre/+}* mice. **b** IF co-localization of Sav1 and ZO-1 in *Scrib^{f/f}*, *Scrib^{f/f}Ltf^{cre/+}*, and *Scrib^{f/f}Pgr^{cre/+}* mice on day 5 afternoon implantation sites. Sav1 expression is absent in *Scrib^{f/f}Pgr^{cre/+}* mice. **c** IF co-localization of Lats1 and ZO-1 in *Scrib^{f/f}*, *Scrib^{f/f}Ltf^{cre/+}*, and *Scrib^{f/f}Pgr^{cre/+}* mice in day 5 afternoon implantation sites. Lats1 expression is decreased in *Scrib^{f/f}Pgr^{cre/+}* mice. **d** IF of YAP in *Scrib^{f/f}*, *Scrib^{f/f}Ltf^{cre/+}*, and *Scrib^{f/f}Pgr^{cre/+}* mice on day 5 afternoon implantation sites. *Scrib^{f/f}Pgr^{cre/+}* implantation site shows increased levels of nuclear YAP in stroma surrounding the embryo. Scale bar: 200 μm. Asterisks indicate the location of embryos. Each image is a representative of at least three independent experiments. M mesometrial pole, AM antimesometrial pole, le luminal epithelium, st stroma, bv blood vessels.

Scrib. We also noticed that the crypt epithelium remains intact in *Scrib^{f/f}Pgr^{cre/+}* mice as demarcated by E-cad staining (Fig. 6c). We have recently shown that the blastocyst trophectoderm escapes from the crypt epithelium to connect with the underlying stroma through a process of entosis[31]. This is followed by apoptosis of the epithelial cells at the bottom of the crypt on day 5 evening as marked by signals of cleaved Caspase 3. The apoptosis signature becomes more prominent on day 6 morning to establish a stronger anchorage of the embryo with the endometrium. Cleaved Caspase 3 signals are absent when implantation is faulty due to defective gland-implantation crypt assembly[2,4]. Indeed, we found that caspase 3 signals are missing at the site of the embryo in *Scrib^{f/f}Pgr^{cre/+}* mice with intact crypt epithelium marked by CK8 immunostaining on day 6 (Supplementary Fig. 5a). Collectively, these results point toward abnormal PDZ formation in *Scrib^{f/f}Pgr^{cre/+}* mice impairing the process of implantation.

The PDZ is comprised of three to five tightly packed epithelial-like cell layers, which is thought to safeguard embryos from harmful maternal circulating substances and immune cells[6,32]. In fact, a PDZ marker ZO-1 is not expressed in *Scrib^{f/f}Pgr^{cre/+}* mice on day 5 afternoon when the PDZ normally begins to form. However, *Scrib^{f/f}Pgr^{cre/+}* mice show ZO-1 expression in stromal cells surrounding the embryo on day 6, suggesting that PDZ formation had been considerably slowed in *Scrib^{f/f}Pgr^{cre/+}* mice (Supplementary Fig. 5b). To show that the PDZ is avascular[6,33], we examined the expression status of Angiopoietin-2 (*Ang-2*), a negative regulator of angiogenesis[34]. Ang2 shows strong expression in the PDZ area of floxed and *Scrib^{f/f}Ltf^{cre/+}* females at the implantation site on the morning of day 6. We found that expression is much lower in *Scrib^{f/f}Pgr^{cre/+}* mice on day 6 compared with floxed and *Scrib^{f/f}Ltf^{cre/+}* mice as examined by fluorescence in situ hybridization (Fig. 6d). In the same context, FLK1, an endothelial cell marker, required for angiogenesis during early pregnancy[34], was evaluated in these mice. Co-staining of FLK1 and ZO-1 shows that the PDZ is indeed an avascular area on day 6 morning in floxed and *Scrib^{f/f}Ltf^{cre/+}* mice, but many FLK1-positive cells with signs of primitive vascular structures were present in the PDZ area in *Scrib^{f/f}Pgr^{cre/+}* mice (Fig. 6e and Supplementary Fig. 6). These results indicate the PDZ formed in *Scrib^{f/f}Pgr^{cre/+}* females is not only morphologically but also functionally defective. These effects were associated with more infiltration of CD45 cells (a common marker of leukocytes) in the PDZ area of *Scrib^{f/f}Pgr^{cre/+}* females with a few of these cells often found situated within the crypt on day 6 (Supplementary Fig. 7).

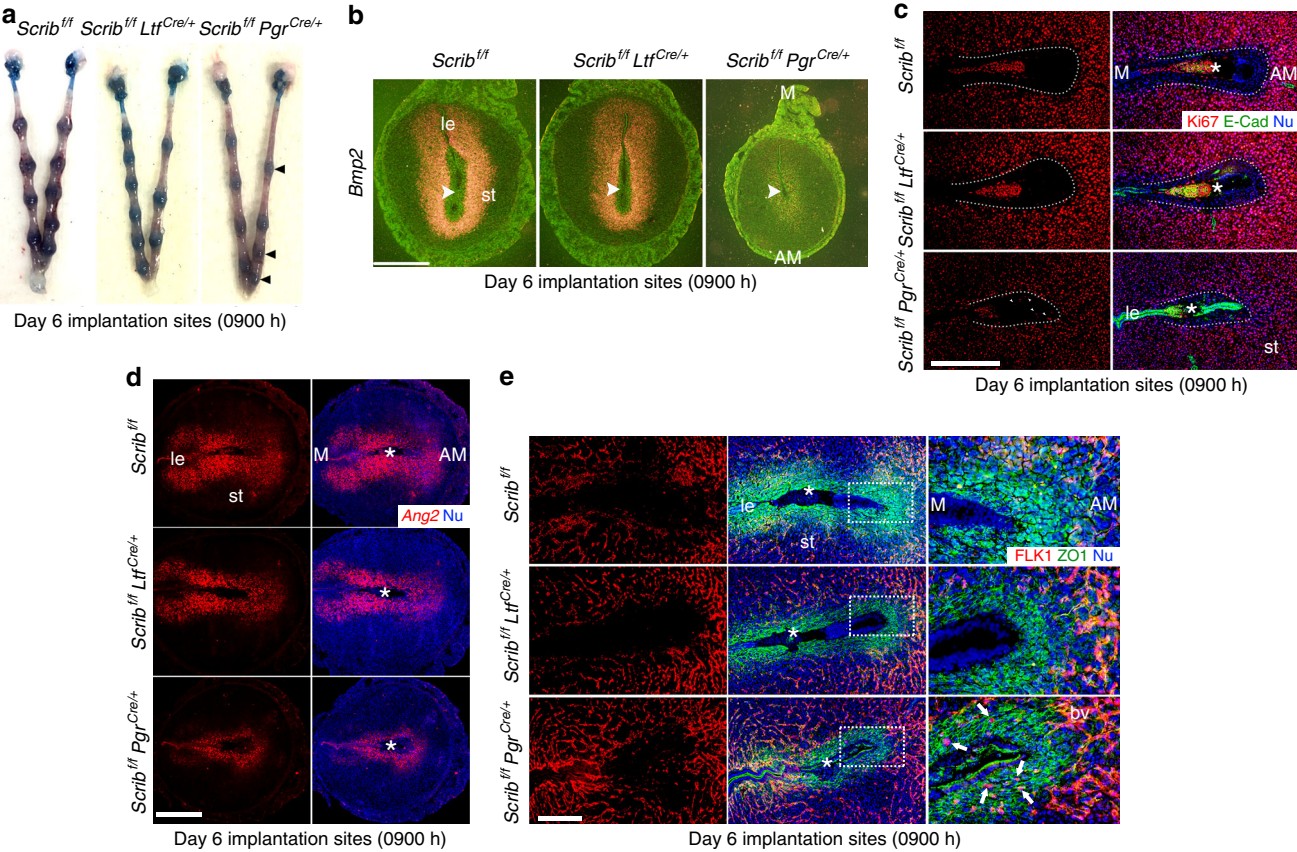

**Fig. 6** *Scrib* deletion in stroma impedes formation of avascular PDZ. **a** Representative images of implantation sites in *Scrib^f/f^*, *Scrib^f/f^Ltf^cre/+^*, and *Scrib^f/f^Pgr^cre/+^* mice on day 6. Arrowheads indicate weak implantation sites. **b** In situ hybridization of *Bmp2* in *Scrib^f/f^*, *Scrib^f/f^Ltf^cre/+^*, and *Scrib^f/f^Pgr^cre/+^* mice implantation sites on day 6. Arrowheads indicate the location of blastocysts. Scale bar: 200 μm. **c** IF of Ki67 and E-cad in *Scrib^f/f^*, *Scrib^f/f^Ltf^cre/+^*, and *Scrib^f/f^Pgr^cre/+^* mice on day 6 implantation sites. Insides of dotted lines indicate PDZ, arrowheads indicate Ki67 positive stromal cells in PDZ area. Scale bar: 400 μm. **d** In situ hybridization of *Ang-2* in *Scrib^f/f^*, *Scrib^f/f^Ltf^cre/+^*, and *Scrib^f/f^Pgr^cre/+^* mice implantation sites on day 6. Scale bar: 500 μm. **e** IF of ZO-1 and FLK-1 in day 6 implantation sites of *Scrib^f/f^*, *Scrib^f/f^Ltf^cre/+^*, and *Scrib^f/f^Pgr^cre/+^* mice show blood vessels invaded the PDZ in *Scrib^f/f^Pgr^cre/+^* mice as indicated by arrows. Scale bar: 200 μm. Asterisks indicate the location of embryos. Each image is a representative of at least three independent experiments. M mesometrial pole, AM antimesometrial pole, le luminal epithelium, st stroma, bv blood vessels.

The distribution of CD45 cells in *Scrib^f/f^* and *Scrib^f/f^Ltf^cre/+^* are comparable (Supplementary Fig. 7). These results further suggest that the PDZ is defective in *Scrib^f/f^Pgr^cre/+^* females.

The formation of an aberrant PDZ in *Scrib^f/f^Pgr^cre/+^* implantation sites on day 6 is associated with much weaker expression of Sav1 and Lats1 (Hippo signaling) in *Scrib^f/f^Pgr^cre/+^* mice (Supplementary Fig. 8a, b), and upstream proteins MST1/MST2 show decreased phosphorylation in day 6 implantation sites of *Scrib^f/f^Pgr^cre/+^* mice. These results suggest that ablation of Scrib in stromal cells compromises Hippo signaling pathways in the PDZ after implantation (Supplementary Fig. 8c). Conversely, with the beginning of SDZ development on day 6, there is clear evidence for YAP expression in the SDZ of *Scrib^f/f^* and *Scrib^f/f^Ltf^cre/+^* mice, but not in *Scrib^f/f^Pgr^cre/+^* mice due to abnormal formation of the PDZ (Supplementary Fig. 8d). The staining of the PDZ by E-cadherin (Fig. 6c), CK8 (Supplementary Fig. 5a), and β-catenin (Supplementary Fig. 8d) suggests minimal or absence of invasion by trophoblasts in *Scrib^f/f^Pgr^cre/+^* uteri[31]. Collectively, these results show that stromal cell Scrib plays a critical role in PDZ formation.

**Defective PDZ formation dysregulates gland-crypt assembly.** The observations in *Scrib^f/f^Pgr^cre/+^* mice led us to conjecture that compromised PDZ formation limits the appropriate cross-talk between the embryo and uterus. We have recently shown that proper crypt-gland assembly in implantation is critical for pregnancy success[2,4]. Using tissue clearing and 3D imaging, we observed that *Scrib^f/f^Pgr^cre/+^* mice do not show correct crypt-gland assembly on day 6 morning (Fig. 7). It is an open question as to how PDZ formation becomes aberrant in *Scrib^f/f^Pgr^cre/+^* mice with deletion of *Scrib* in the sub-epithelial stroma, but not in *Scrib^f/f^Ltf^cre/+^* mice, which have presumably a normal PDZ with appropriate gland elongation and crypt-gland assembly (Fig. 7). Our results suggest that stromal cell Scrib is required to form and establish the PDZ with epithelioid-like cells surrounding the blastocyst, since epithelial Scrib expression normally disappears in the crypt at this time.

## Discussion

The quality of implantation is a critical determinant for pregnancy success. The major events in pregnancy include uterine receptivity, implantation, decidualization, placentation, and parturition. Early pregnancy events are further divided into subcategories: uterine receptivity, epithelial remodeling to form the crypt (implantation chambers and crypt-glands assembly) and epithelial-stromal cross-talk. These events are followed by decidualization (primary and secondary decidualization).

Appropriate PCP signaling involving Wnt5a–ROR–Vangl2 is critical to crypt formation for embryo implantation[3,4]. In this context, epithelial deletion of *Vangl2*, a core PCP component,

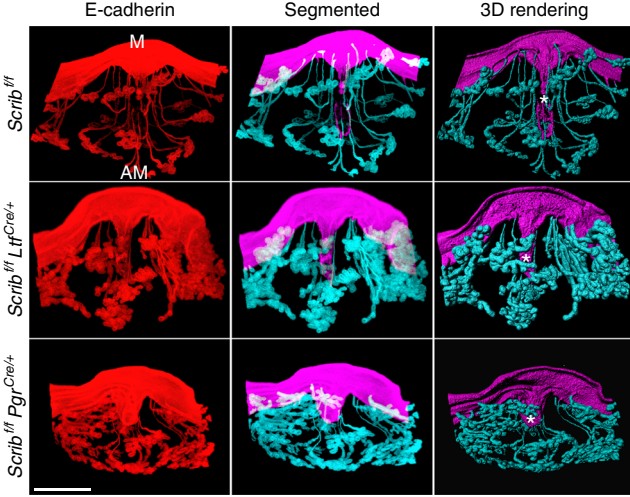

**Fig. 7** 3D imaging of day 6 implantation sites in *Scrib*<sup>f/f</sup>*Pgr*<sup>cre/+</sup> females shows inferior crypt-gland assembly. Images of E-cadherin immunostaining, segmented and 3D rendered images of day 6 implantation sites in each genotype show deficient development of gland-implantation epithelial structure in *Scrib*<sup>f/f</sup>*Pgr*<sup>cre/+</sup> females. Images were generated by a Nikon A1R Multiphoton Microscope with LWD 10× objective with 7 μm Z-stack. Scale bar, 500 μm. Asterisks indicate the location of embryos. M mesometrial pole, AM antimesometrial pole. Each image is a representative of at least three independent experiments.

results in defective implantation and compromised decidualization[2]. Scrib directly and physically interacts with Vangl2[8], and conditional deletion of *Vangl2* shows abnormal Scrib localization and expression in the uterus during implantation[2,4]. Our observations of marginal adverse effects on pregnancy success after epithelial deletion of *Scrib* by an *Ltf-Cre* driver, as opposed to severe subfertility with the deletion of stromal *Scrib* by *Pgr-Cre*, suggest that Scrib has an important and unique role in the stromal compartment in early pregnancy. Scrib is normally absent in the crypt epithelium following the blastocyst attachment. These results suggest that stromal Scrib is essential for PDZ formation to sustain subsequent pregnancy events.

Pregnancy phenotypes seen in mice with *Scrib* deletion in the epithelium and/or the stroma are distinct from those in mice with epithelial deletion of *Vangl2* or *Scrib* deletion in both the epithelium and stroma by *Ltf-Cre* and *Pgr-Cre* drivers, respectively. In this respect, *Vangl2*<sup>f/f</sup>*Ltf*<sup>cre/+</sup> mice or *Vangl2*<sup>f/f</sup>*Pgr*<sup>cre/+</sup> mice show substantial compromised pregnancy outcomes[2,4]. In contrast, the apparently normal pregnancy outcomes in *Scrib*<sup>f/f</sup>*Ltf*<sup>cre/+</sup> females, but not in *Scrib*<sup>f/f</sup>*Pgr*<sup>cre/+</sup> mice, clearly illuminates the similar and distinct roles of Scrib versus Vangl2 in uterine biology. It may be argued that epithelial Scrib is dispensable, since Scrib expression in the crypt epithelium is downregulated with the initiation of implantation along with increased expression in the underlying stroma. It is possible that the loss of *Scrib* in the epithelium is compensated by Vangl2, a major PCP component[4]. The alternative scenario is that stromal Scrib alone is critical to form and shape the PDZ. This possibility is more reasonable, since *Scrib* deletion in the epithelium is associated with higher Scrib expression in the underlying sub-epithelial stroma after embryo attachment on day 5.

Although epithelial–mesenchymal interaction is important for uterine biology and pregnancy success[1], whether stromal cells participate in this interaction with the epithelium to form a crypt-gland assembly is not yet known. The proliferation and differentiation of stromal cells accompanied by increased angiogenesis

is critical for decidualization; normally the implanting embryo is the stimulus for triggering decidualization in the receptive uterus. As stated above, stromal cells close to the implantation chamber undergo differentiation to an epithelial-like tight junction permeability barrier that forms the PDZ on day 5[21]. This avascular zone becomes fully established on day 6[6,33]. In this case, it seems that PDZ formation is analogous to a transitory stromal-epithelial transition acquiring avascular status and is a key event in pregnancy, since compromised PDZ formation leads to pregnancy failure and infertility[1]. The PDZ is surrounded by proliferating and differentiating stromal cells termed the SDZ, which peaks on day 8. Because of the epithelial-like characteristics of the PDZ, the function of this zone is considered to provide a protective barrier to embryos from maternal immune surveillance or other noxious materials[6,35]. With the expansion of the SDZ, the PDZ undergoes demise, but how this occurs is not clearly understood at this time.

Floxed or *Scrib*<sup>f/f</sup>*Ltf*<sup>cre/+</sup> females show normal avascular PDZ formation and expression of decidua-specific gene expression as opposed to *Scrib*<sup>f/f</sup>*Pgr*<sup>cre/+</sup> mice, which show compromised PDZ formation beginning on day 5 of pregnancy with low expression of *Bmp2*, *Wnt4*, and *Hoxa10* in stromal cells dispersed with blood vessels into the designated area of the PDZ on day 6. One potential role of epithelioid PDZ cells is to serve as "guard cells" to gradually transition the embryo from the hypoxic milieu within the luminal and crypt epithelium to the normoxic environment with the progression of pregnancy, as a way of averting sudden oxidative stress to the embryo. Indeed, early stage embryos and embryonic stem cells (ES) show superior growth under lower $O_2$ tension than those grown in higher $O_2$ environments[36–40]. $O_2$ tension within the oviduct and uterine lumen is hypoxic compared with higher $O_2$ tension in the stroma[36]. The establishment of the avascular and transient epithelioid PDZ in the stromal bed fulfills this task.

Aberrant ZO-1 and Hippo signaling with deletion of *Scrib* in the stroma surrounding the implanting blastocyst in *Scrib*<sup>f/f</sup>*Pgr*<sup>cre/+</sup> mice suggest that faulty PDZ formation is due to aberration of these signaling pathways (Supplementary Fig. 9). Although Scrib is clearly expressed in the epithelium in early pregnancy stages, its role in cell polarity and cell adhesion appears dispensable with regards to pregnancy success. However, Scrib expression rapidly increases in stromal cells around the embryo after its attachment and is critical for PDZ initiation and formation, which contributes to crypt-glands assembly that leads to pregnancy success. Our findings reveal for the first time a role of stromal cell Scrib to initiate and form the PDZ for pregnancy success. Whether the PDZ is formed during implantation in humans and other subhuman primates is not known, though the epithelial plaque formed during early pregnancy in macaque may have a similar function[41].

## Methods

**Mice**. P*gr*<sup>cre/+</sup> and *Ltf*<sup>Cre/+</sup> mouse lines were generated as described before[11,12]. *Scribble*-floxed mouse line (*Scrib*<sup>f/f</sup>) was originally generated by Tobias B. Huber's laboratory[42]. *Scrib*<sup>f/f</sup>*Pgr*<sup>cre/+</sup> and *Scrib*<sup>f/f</sup>*Ltf*<sup>cre/+</sup> mice were generated by mating *Scrib*<sup>f/f</sup> females with *Pgr*<sup>cre/+</sup> and *Ltf*<sup>cre/+</sup> males, respectively. All mice used in this study were housed in Cincinnati Children's hospital animal care facility with constant 12 h/12h-light/dark cycle following NIH and institutional guide lines for the animal care and use committee. Mice were provided with autoclaved laboratory rodent diet 5010 (purina) and UV light-sterilized reverse osmosis/deionized constant circulation water ad libitum. At least three mice were used for every individual experiment in each mouse model.

**Analysis of pregnancy events**. Pregnancy events were analysed as described previously[2,4,14]. Briefly, three adult females from each genotype were housed together with a fertile WT male overnight in separate cages. The morning of finding the vaginal plug was considered day 1 of pregnancy. Plug-positive females were kept separately for pregnant experiments. Litter size, pregnancy rate, and outcomes were monitored for the whole pregnancy process. To confirm that plug-positive mice were pregnant on day 4 of pregnancy, one uterine horn was flushed with saline to detect blastocyst existence. For day 5 and day 6 pregnant mice,

100 μL of 1% Chicago blue in saline was injected intravenously to visualize implantation sites as blue bands[43]; if no blue band was observed, uterine horns were flushed to check for the presence of embryos.

**Histology**. Tissue sections from control and experimental groups were processed on the same slide. Frozen sections (12 μm) were fixed in 4% PFA-PBS for 10 min at room temperature and then stained with hematoxylin and eosin for light microscopy analysis.

**In situ hybridization**. The frozen tissue pieces (5 mm long) were mounted on the specimen stage with a small amount of semi-frozen OCT compound (Sakura) to hold the tissue vertically without completely covering the tissue. The tissue blocks were kept within the cryostat until sectioning. Sections were mounted on test slides and checked under the microscope to find embryo implantation sites. Frozen sections (12 μm) from each genotype were processed onto the same slides. In situ hybridization using $^{35}$S-labeled probes were performed as previously described[14,44]. Signals were visualized under a Nikon Eclipse E800 with dark-field. Fluorescence in situ hybridization was adopted based on previously established DIG in situ hybridization[4]. In brief, following proteinase K (5 μg/ml) digestion and acetylation, slides were hybridized at 55 °C with the DIG-labeled Ang2 probe[34]. Anti-Dig-peroxidase was applied onto hybridized slides following washing and peroxide quenching. Color was developed by TSA (Tyramide signal amplification) Fluorescein according to the manufacturer's instructions (PerkinElmer).

**Immunofluorescence (IF)**. The frozen tissues were mounted on the specimen stages in the same way as described for "in situ hybridization". Sections with 12 μm thickness were mounted on a poly-l-lysine coated slide. Sections were fixed in cold 4% paraformaldehyde for 10 min after air drying. Slides were then processed for IF staining. IF was performed as previously described[2,4]. Frozen sections (12 μm) from each genotype were processed onto the same slides and incubated with primary antibodies listed in Supplementary Table 2. For signal detection, secondary antibodies listed in Supplementary Table 2 were used. Nuclear staining was performed using Hoechst 33342 (5 μg/mL, H1399, Thermo Scientific). Pictures were taken using the Nikon Eclipse 90i upright microscope and processed by Nikon Elements Viewer.

**Whole-mount immunostaining for 3D imaging**. Samples were fixed in Dent's Fixative (Methanol:DMSO (4:1)) overnight at −20 °C. After fixation, tissues were then washed in 100% Methanol three times for 1 h each and bleached with 3% $H_2O_2$ in methanol overnight at 4 °C to eliminate pigmentation. The samples were washed in PBS-T containing 0.1% Tween20 three times for 1 h each at room temperature and then blocked in 5% BSA in PBS-T overnight at 4 °C. The samples were then incubated with anti-E-cadherin antibody (1:100, 3195 s, Cell Signaling Technology) on a rotor for 7 days at 4 °C. After incubation, the samples were then washed in PBS-T 6 times for 1 h each at room temperature and then incubated with Alexa Fluor® 594 AffiniPure Donkey Anti-Rabbit IgG (H + L) (1:300, Jackson ImmunoResearch) on a rotor for 4 days in a light proof tube at 4 °C. The samples were stored in the dark until tissue clearing.

**Tissue clearing for 3D imaging**. The stained samples were held straight with forceps in 100% methanol for 1 min to align the mesometrial–antimesometrial (M–AM) axis and then dehydrated in 100% methanol for 30 min. Dehydration was followed by tissue clearing by BABB (Benzyl alcohol: Benzyl benzoate (1:2); each reagent from Sigma-Aldrich) for 1 h at room temperature. The samples were then stored in the dark until 3D imaging acquisition.

**3D imaging and processing**. 3D pictures were acquired by a Nikon multiphoton upright confocal microscope (Nikon A1R) with LWD 16X water objective with 3 μm Z-stack. To obtain the 3D structure of the tissue, the surface tool Imaris (version 9.2.0., Bitplane) was used.

**Quantitative RT-PCR**. RNA isolation and qRT-PCR were performed as described[14,44] using the following primers: 5′-CTGCGTCGCTGTCTTTCCT-3′ and 5′-TTCGGTCTAACCATAACTCCC-3′ for Scrib; 5′-GCAGATGT ACCGCACTGAGATTC-3′ and 5′-ACCTTTGGGCTTACTCCATTGATA-3′ for Rpl7; Rpl7 served as an internal control.

**Western blotting (WB)**. WB was performed as described[14,44]. Primary antibodies listed in Supplementary Table 2 were used. β-Actin was used as a loading control. For signal detection, blots were incubated with secondary antibodies listed in Supplementary Table 2 followed by the incubation with Clarity™ Western ECL Substrate (Bio-Rad). Bands were visualized under Amersham Imager 680 (GE Healthcare) or X-OMAT 2000 (Kodak). Uncropped scans of each western blot are shown in Supplementary Figs. 10–12.

**Statistics**. The data were analyzed by One-way ANOVA followed by Bonferroni post-hoc test. $P < 0.05$ was considered statistically significant.

**Reporting summary**. Further information on research design is available in the Nature Research Reporting Summary linked to this article.

## Data availability
The authors declare that all data supporting the findings of this study are available within the article and its Supplementary Information files or from the corresponding authors on reasonable request.

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

## Acknowledgements

Our sincere thanks to Katie Gerhardt for her efficient editing of the manuscript. We are grateful to Mireille Montcouquiol (France) for sharing with us a specific antibody for Scrib. This work was supported in part by NIH grants (HD068524 and DA006668 to S.K. D) and a March of Dimes Centre grant (22-FY17-889). S.A. was supported by Astellas Foundation for Research on Metabolic Disorder Fellowship and The Osamu Hasashi Memorial Foundation Fellowship for Study Abroad. She is now supported by a JSPS Overseas Research Fellowship. G.W. and F.G. were supported by the CRC 1140. TBH was supported by the DFG (CRC1192, CRC1140, CRC 992), the BMBF (01GM1518C), the European Research Council-ERC grant 616891 and by the H2020-IMI2 consortium BEAT-DKD.

## Author contributions

J.Y., S.A., W.D., A.B., and X.S. performed experiments, J.Y. and S.K.D. designed experiments. J.Y., S.A., W.D., and S.K.D. analysed data. J.Y., S.A., and S.K.D wrote the manuscript. G.W., F.G., and T.H. designed and developed the conditional mouse model.

## Competing interests

The authors declare no competing interests.
