## [Peer Review File · Nature Communications]

Reviewers' comments:

Reviewer #1 (Remarks to the Author):

SUMMARY

This manuscript focuses on planar cell polarity signaling (PCP) in the uterus during pregnancy establishment. Specifically, the manuscript is focused on a PCP component scaffold protein Scribble (Scrib). The authors found that uterine deletion of Scrib by a Pgr-Cre driver leads to defective formation of the primary decidual zone (PdZ) and formation of the implantation chamber (crypt), thereby compromising pregnancy outcomes. Of note, deletion of Scrib using the epithelial-specific Ltf-iCre driver had no effect on pregnancy. The study provides new information on how the decidua is formed during pregnancy, which is critical for pregnancy establishment and involved in the etiology of later pregnancy complications and miscarriage.

OVERALL AND MAJOR COMMENTS

Overall, this manuscript contains a series of studies on Scrib conditional knockout (cKO) mice that are very well executed with appropriate discussion of the results and findings. The exception is that the authors do not address why Pgr-Cre: Scrib cKO mice have only a reduction in litter size as opposed to a complete loss of all embryos. What are the proposed mechanisms by which some embryos survive to term in the Pgr-Cre: Scrib cKO mice?

Major Comments:

- (1) Was there an effect of parity on litter size in Pgr-Cre: Scrib cKO mice?
- (2) There are numerous instances of words not spelled correctly.
- (3) How do the authors know that they are evaluating a good versus a bad embryo during early pregnancy (days 4, 5 and 6) given that half or more of the embryos are lost by day 12 of pregnancy?
- (4) Is Scrib upregulated in human endometrial stromal cells during in vitro decidualization or in the secretory as compared to proliferative phase?

SPECIFIC COMMENTS

Line Comment

- 49 Please cite a review or two concerning ripple effects of implantation failure
- 112 litter is not spelled correctly
- 122 PR should be PGR
- 406 labeled should be labelled

In conclusion, although the studies are well conducted, the phenotype of the Pgr-Cre: Scrib conditional knockout (cKO) mice is not very strong, i.e. the authors report only a 57% reduction in litter size (or embryo loss). The rationale or insights into why a complete loss of all embryos during pregnancy is not addressed in the manuscript. This is a bit troublesome, as how do the authors know they are evaluating a good versus a bad embryo during early pregnancy (days 4, 5 and 6) given that half or more of the embryos are lost by day 12 of pregnancy. This aspect of the manuscript reduces overall enthusiasm for the findings and publication.

Reviewer #2 (Remarks to the Author):

Interesting paper and suitable for the journal but the phenotype in my opinion needs a little more characterisation.

What is the effect of the KO on intercellular junctions in the PDZ? With disruption of ZO-1 tight junctions may be absent but this is not shown. Is there an effect on adherens junctions -- they often stabilise TJs? Is the paracellular space expanded? Are CD45+ cells present in the PDZ? Can further analysis be done of the endothelial cells that are claimed to appear in the subepithelial stroma -- is there any vascular structure? Images in the pdf were not of sufficiently high quality to see in detail.

One might anticipate an effect on trophoblast primary giant cells. For example, do they invade further?

Point-by-point responses to Reviewers' comments

We are pleased with the reviewers' positive notes as well as critical but constructive comments to strengthen our manuscript. Our responses are marked by green font. New additions and changes in the main text are also marked by green font.

Reviewer #1 (Remarks to the Author):

SUMMARY

This manuscript focuses on planar cell polarity signaling (PCP) in the uterus during pregnancy establishment. Specifically, the manuscript is focused on a PCP component scaffold protein Scribble (Scrib). The authors found that uterine deletion of Scrib by a Pgr-Cre driver leads to defective formation of the primary decidual zone (PdZ) and formation of the implantation chamber (crypt), thereby compromising pregnancy outcomes. Of note, deletion of Scrib using the epithelial-specific Ltf-iCre driver had no effect on pregnancy. The study provides new information on how the decidua is formed during pregnancy, which is critical for pregnancy establishment and involved in the etiology of later pregnancy complications and miscarriage.

OVERALL AND MAJOR COMMENTS

Overall, this manuscript contains a series of studies on Scrib conditional knockout (cKO) mice that are very well executed with appropriate discussion of the results and findings. The exception is that the authors do not address why Pgr-Cre:Scrib cKO mice have only a reduction in litter size as opposed to a complete loss of all embryos. What are the proposed mechanisms by which some embryos survive to term in the Pgr-Cre:Scrib cKO mice?

Responses to Reviewer 1

We are pleased with his/her thoughtful but critical comments.

1. In this study, we utilized a *Pgr-Cre* driver to delete *Scrib* in the uterine cells, which express progesterone receptors (PGR). Since early embryos do not express functional PGR, we do not expect deletion of *Scrib* in embryonic cells. A large number of studies on uterine functions, including by our lab, have used the *Pgr-Cre* driver for uterine deletion of genes of interest. Notably, systemic deletion of *Scrib* results in embryonic lethality (**J Clin Invest.** 2011 Nov; 121(11):4257-67).
2. Reviewer 1's concern that *Scrib^{ff}Pgr^{Cre/+}* females have only 57% reduction in litter size is perhaps a misunderstanding of our results. The pregnancy phenotype of *Scrib^{ff}Pgr^{Cre/+}* mice is blatant and strong. The results show that 57% of plug-positive *Scrib^{ff}Pgr^{Cre/+}* females failed to give birth to any pups; the remaining 43% of females that did give birth showed about 50% reduction in litter sizes. The pregnancy phenotypes recorded from a large number of cohort of deleted females over many months assured the validity of the results. This misunderstanding apparently resulted from our unclear phrasing of the results. The results are now presented in **Figure 1** and in a Table (**Supplementary Table 1**). We have also clarified the description of this phenotype in the text of the revised manuscript. It is well-known that Cre expression is not 100% effective and shows biological and/or random variations from animal to animal.
3. We have addressed the potential mechanism by which uterine *Scrib* regulates female fertility and provide evidence that *Scrib* via interactions with ZO-1 and Hippo signaling participates in PDZ formation. The defective PDZ formation in *Scrib^{ff}Pgr^{Cre/+}* mice gives rise to smaller implantation sites and embryo resorptions through adverse ripple effects.

Major Comments:

(1) Was there an effect of parity on litter size in *Pgr-Cre:Scrib* cKO mice?

- Yes, litter sizes did vary (1 to 8 with a median of 3 to 5) in *Scrib^{ff}Pgr^{Cre/+}* females versus litter sizes in *Scrib^{ff}* (control) females (7 to 10). We will clearly

present these results in the revised manuscript. It is not surprising that biological variations are more pronounced in conditionally gene-deleted animals.

(2) *There are numerous instances of words not spelled correctly.*

We have corrected the spelling errors in the revised manuscript.

(3) *How do the authors know that they are evaluating a good versus a bad embryo during early pregnancy (days 4, 5 and 6) given that half or more of the embryos are lost by day 12 of pregnancy?*

The *Pgr-Cre* driver is capable of deleting genes of interest in uterine cells expressing PGR, but it does not affect preimplantation embryos since they are not known to express functional PGR. There are numerous reports of the use of the *Pgr-Cre* driver to delete genes in uterine cells without affecting preimplantation embryo development.

However, we cannot completely exclude the possibility that altered uterine milieu in *Scrib^{fl/fl}Pgr^{Cre/+}* mice in turn affects early embryo development. That is why we carefully examined the progression of pregnancy with respect to uterine changes and embryonic development. First, we examined the expression of uterine cell proliferation markers, estrogen and progesterone receptors (ESR1 and PGR) on day 4 of pregnancy. To do so, one uterine horn was flushed to recover blastocysts to confirm pregnancy, and the contralateral horn was used for expression studies. Comparable expression patterns along with normal uterine histological appearance between the deleted and control uteri suggest that uterine receptivity was unaffected and blastocysts appeared morphologically normal in *Scrib^{fl/fl}Pgr^{Cre/+}* females. Furthermore, expressions of blastocyst attachment markers *Ptgs2* and *Hbegf* are comparable among the genotypes. Notably, deletion of epithelial *Scrib* by the *Ltf-Cre* have little adverse effects on pregnancy outcomes. Adverse effects of uterine *Scrib* deletion by a *Pgr-Cre* driver are seen heralding the beginning of PDZ formation on day 5. On day 8, embryonic demise was evident with continued embryonic resorptions

seen on day 12 due to adverse ripple effects. The concept of “adverse ripple effects” was reported first by us and confirmed by others using different gene-deleted and wild-type mouse models (reviewed in *Nat Med* 18(12): 1754-1767, 2012).

(4) Is Scrib upregulated in human endometrial stromal cells during in vitro decidualization or in the secretory as compared to proliferative phase?

It is not expected that Scrib will be upregulated in human decidual cells in vitro or in the secretory phase, since it is expressed at a particular stage of implantation, especially in stromal cells transitioning to epithelioid cells to form PDZ in mice. In this respect, we still do not know the precise timing of implantation in humans, precluding research on PDZ formation. Nonetheless, we examined *in vitro* decidualized cells derived from human uterine fibroblasts (HuF). This cell line was developed by the late Stuart Handwerger in our institute and has been used in many studies (**Endocrinology** 2011;152(11):4368–4376; **J Clin Invest** 2016 Aug 1; 126(8):2941-54). HuF cells decidualized *in vitro* express marker genes prolactin and IGFBP1. Upon examining HuF cells undergone *in vitro* decidualization, we did not see any significant changes in Scrib expression by qPCR, although decidual marker genes were upregulated; the possible reason is that *in vitro* decidualized stromal cells morphologically are not epithelial-like cells. We include these data for Reviewer 1 for his/her perusal, not for inclusion in the manuscript (see below).

The relative expressions of *SCRIB*, *PRL* and *IGFBP1* in HuF cells (a human endometrial cell line) after decidual stimuli. Cells were treated with 1 μM P_4 , 10 nM E_2 and 1 μM PGE_2 to induce decidualization. After 6 days culture, cells were collected for RNA extractions. *GAPDH* was used as an internal control, *PRL* and *IGFBP1* were used as decidualization markers. $n = 3$ for each group, Data are mean \pm SEM. * $P < 0.05$ and *** $P < 0.001$ by student' s t-test.

SPECIFIC COMMENTS

Line Comment

49 Please cite a review or two concerning ripple effects of implantation failure

- We cited one review article and two original publications.

112 litter is not spelled correctly

- This has been corrected.

122 PR should be PGR

- PR has been changed to PGR

406 labeled should be labelled

- This has been corrected to “labelled”.

In conclusion, although the studies are well conducted, the phenotype of the Pgr-Cre:Scrb conditional knockout (cKO) mice is not very strong, i.e. the authors report only a 57% reduction in litter size (or embryo loss). The rationale or insights into why a complete loss of all embryos during pregnancy is not addressed in the manuscript. This is a bit troublesome, as how do the authors know they are evaluating a good versus a bad embryo during early pregnancy (days 4, 5 and 6) given that half or more of the embryos are lost by day 12 of pregnancy. This aspect of the manuscript reduces overall enthusiasm for the findings and publication.

- This concern has already been addressed above. We believe this concern is the result of misunderstanding; perhaps we failed to clearly present our data.

-

Reviewer #2 (Remarks to the Author)

Responses to Reviewer 2

We are very thankful to this reviewer for his/her positive comments.

Interesting paper and suitable for the journal but the phenotype in my opinion needs a little more characterization.

We have provided additional characterization of the phenotype in the revised version.

- *What is the effect of the KO on intercellular junctions in the PDZ? With disruption of ZO-1 tight junctions may be absent but this is not shown.*

- One of the established roles of Scrib is maintenance of junctional networks. Importantly, Scrib has been shown to bind ZO-1 (**Am J Pathol** 2010, Jan;176(1):134-45.). Deletion or down-regulation of Scrib results in impaired cell-cell junctions (**J Cell Biol** 2019 Mar 4;218(3):742-756). Therefore, we believe that Scrib stabilizes tight junctions by ZO-1 and contributes to the formation of PDZs. This is now described in the text of the revised version.

Is there an effect on adherens junctions -- they often stabilise TJs?

- As shown in Sup. Fig. 8d, we did immunofluorescence staining of β -catenin, a marker of adherens junctions (AJs). Similar to ZO-1 and Scrib, β -catenin signaling is also reduced in the PDZ in *Scrib^{fl/fl}Pgr^{Cre/+}* uteri. Since there are reports that Scrib can bind to AJ proteins including β -catenin (**Proc Natl Acad Sci USA** 2014 Feb 18;111(7):2542-7.; **J Biol Chem** 2006 Aug 4;281(31):22299-311.), it is conceivable that Scrib supports the formation of AJs as well as tight junctions.

Is the paracellular space expanded?

- We do not know. We do know that paracellular space helps in transporting substances across the epithelium through the intercellular space between cells. Since the PDZ is comprised of a few layers of epithelioid cells that permit selective permeability, its disruption in *Scrib^{fl/fl}Pgr^{Cre/+}* females is likely to increase the paracellular space. This is also evident from the expansion of blood vessels and entry of more CD45 cells in *Scrib^{fl/fl}Pgr^{Cre/+}* PDZs.

Are CD45+ cells present in the PDZ?

- Normally CD45 cells are rare in the PDZ, but we see a few more CD45 positive cells in the PDZ area in *Scrib^{fl/fl}Pgr^{Cre/+}* females on day 6 (**Supplementary Figure 7**)

Can further analysis be done of the endothelial cells that are claimed to appear in the subepithelial stroma -- is there any vascular structure? Images in the pdf were not of sufficiently high quality to see in detail.

- We have provided a higher magnification of Flk1 staining, which shows more clearly the beginning of the formation of vascular structures (**see supplementary Figure 6**).

One might anticipate an effect on trophoblast primary giant cells. For example, do they invade further?

- The staining of the PDZ by E-cadherin, β -catenin and CK8 suggests minimal or absence of invasion by trophoblasts.

We hope that our responses are satisfactory to the reviewers and to the editor.

Sincerely,
SK Dey

REVIEWERS' COMMENTS:

Reviewer #1 (Remarks to the Author):

OVERALL COMMENTS

The authors have very comprehensively and carefully addressed the reviewer comments from the initial review. As such, the manuscript is considerably improved.

MAJOR COMMENTS

(1) The text and figures should consistently utilize appropriate nomenclature for genes. For instance, PR is Pgr and ERalpha is Esr1.

(2) The description of the tissue clearing and 3D visualization of the uterus should be explained in more detail for the sake of reproducibility. This detail on the tissue clearing, antibody infiltration, imaging, and 3D visualization could be provided in the Supplementary Information with great detail. The description in the cited paper (Yuan et al., Nature Comm 2018) is not sufficient in terms of whole mount immunostaining, tissue clearing, and 3D imaging, processing and analysis.

SPECIFIC COMMENTS

Line Comment

76 Provide a reference the previous study

411 The precise method of how the uterus was embedded for cryosectioning and processed thereafter is important for rigor and reproducibility.

416 labeled should be labeled

425 A supplementary table should be provided with the description of all antibodies used for IF and the conditions of the secondary antibodies used for detection to help improve rigor and reproducibility.

Fig. 1c & 1d Day of pregnant should by "Day of pregnancy" and delete uteri

Reviewer #2 (Remarks to the Author):

The manuscript is improved with the responses to both referees.

> Line 109,110: 'although distinct expression of Scrib and its role in PCP is noted in the floxed epithelia'

I do not see the point of this statement as the text above explains that stromal scrib is retained in the UE-specific KO.

>from rebuttal: '... we believe that Scrib stabilizes tight junctions by ZO-1 and contributes to the formation of PDZs. This is now described in the text of the revised version.'

I do not see this in the amended text.

>from rebuttal: 'The staining of the PDZ by E-cadherin, β -catenin and CK8 suggests minimal or absence of invasion by trophoblasts.'

This important point should be added to the results section.

Responses to the reviewers' comments

We express our sincere gratitude to the reviewers for their continued attempts to strengthen the manuscript. Our responses (green fonts) follow the reviewers' comments.

REVIEWERS' COMMENTS:

Reviewer #1 (Remarks to the Author):

OVERALL COMMENTS

The authors have very comprehensively and carefully addressed the reviewer comments from the initial review. As such, the manuscript is considerably improved.

We thank this reviewer for his/her positive statement.

MAJOR COMMENTS

(1) The text and figures should consistently utilize appropriate nomenclature for genes. For instance, PR is Pgr and ERalpha is Esr1.

We have made sincere efforts to be consistent with the gene names throughout the manuscript.

(2) The description of the tissue clearing and 3D visualization of the uterus should be explained in more detail for the sake of reproducibility. This detail on the tissue clearing, antibody infiltration, imaging, and 3D visualization could be provided in the Supplementary Information with great detail. The description in the cited paper (Yuan et al., Nature Comm 2018) is not sufficient in terms of whole mount immunostaining, tissue clearing, and 3D imaging, processing and analysis.

We have now provided a detailed protocol for 3D visualization in the Methods section.

SPECIFIC COMMENTS

Line Comment

76 Provide a reference the previous study

A reference is now provided.

411 The precise method of how the uterus was embedded for cryosectioning and processed thereafter is important for rigor and reproducibility.

Details of the methods are now clearly shown in the text in the Methods section (Page 20-21).

416 labeled should be labeled

“labeled” has been corrected to “labeled”.

425 A supplementary table should be provided with the description of all antibodies used for IF and the conditions of the secondary antibodies used for detection to help improve rigor and reproducibility.

A table giving antibody descriptions is now provided.

Fig. 1c & 1d Day of pregnant should be “Day of pregnancy” and delete uteri

We have changed to “Day 12 of pregnancy” and “Day 8 of pregnancy”.

Reviewer #2 (Remarks to the Author):

The manuscript is improved with the responses to both referees.

We appreciate the encouraging comments of this reviewer.

> Line 109,110: 'although distinct expression of Scrib and its role in PCP is noted in the floxed epithelia'

I do not see the point of this statement as the text above explains that stromal scrib is retained in the UE-specific KO.

This statement is now excluded.

>from rebuttal: '... we believe that Scrib stabilizes tight junctions by ZO-1 and contributes to the formation of PDZs. This is now described in the text of the revised version.'

I do not see this in the amended text.

Sorry for this omission. This is now stated in the Text (Page 11).

>from rebuttal: 'The staining of the PDZ by E-cadherin, β -catenin and CK8 suggests minimal or absence of invasion by trophoblasts.'

This important point should be added to the results section.

This statement is now added (Page 15).